# DAE: DIFFUSION-ALIGNED EMBEDDINGS

## ABSTRACT

This paper introduced DAE, which formulates dimensionality reduction as aligning diffusion processes between high- and low-dimensional spaces. By minimizing the Path-KL divergence—which uniquely captures both transition probabilities and waiting times of continuous-time random walks—we proved formal bounds on generator and semigroup closeness, guaranteeing structure preservation across scales. Our optimization algorithm decomposes this objective into attraction-repulsion terms with an unbiased gradient estimator, enabling efficient parallel implementation. Experiments on single-cell RNA-seq datasets showed DAE consistently preserves both local neighborhoods and global structure, while our CUDA implementation scales to millions of cells with competitive runtime.

The Path-KL framework provides theoretical guarantees that complement existing diffusion-based methods. DAE will be made available with CPU and GPU implementations.

## 1 INTRODUCTION

Scientists often rely on visualizations of dimension-reduced data to interpret high-dimensional measurements. In single-cell biology, for example, two-dimensional embeddings are routinely employed to validate clusters, assess batch correction, and illustrate developmental continua (Becht et al., 2019; Lopez et al., 2018; La Manno et al., 2018), while in neuroscience they help characterize neural population geometry and brain-state dynamics (Churchland et al., 2012; Cunningham & Yu, 2014; Nieh et al., 2021). While undoubtedly useful, such visualizations can also fabricate clusters—either by fracturing continuous processes or collapsing distinct populations—obscuring genuine relationships (Kobak & Berens, 2019; Luecken & Theis, 2019; Huang et al., 2022) and misguiding scientific inferences (Stringer et al., 2019; Ribeiro de Paula et al., 2021).

Reflecting these challenges, an established paradigm in manifold learning is to model the data as a graph (Tenenbaum et al., 2000; Roweis & Saul, 2000; Belkin & Niyogi, 2003; van der Maaten & Hinton, 2008; McInnes et al., 2018). Nodes represent data points while edges quantify pairwise similarities, encoding both local neighborhoods and their global arrangement. Among graph methods, diffusion-based approaches offer a principled way to model connectivity through random walks on the graph, defining similarity via diffusion processes (Coifman & Lafon, 2006; Moon et al., 2019).

To illustrate this idea, imagine an ant wandering through the data graph, stepping from node to node with probabilities determined by the nodes' similarity.[1] Over short durations, the ant repeatedly revisits nearby nodes, reflecting local structure. Over longer durations, its walk spreads into denser regions and eventually traverses the routes connecting distant parts of the graph, reflecting global structure. For an embedding to be faithful, the ant should have the same experience whether navigating the original high-dimensional graph or the graph induced by the low-dimensional embedding.

A limitation of existing diffusion-based methods is that they require fixing a single diffusion timescale $t$—determining how long the ant is allowed to proceed. For too small values of $t$, the resulting embedding exaggerates local neighborhoods and misses global trends; for too large values, the walk converges to its stationary distribution and collapses meaningful geometry. This creates a central challenge: embeddings are often biased toward either local or global structure.

Rather than committing to a fixed $t$, we aim to preserve diffusion behavior across *all* timescales. To this end, we define alignment along three key dimensions: (i) which nodes are visited, (ii) how often

---

[1]In the spirit of the "ant on a garden hose" metaphor, first noted (to our knowledge) by Duff (1984).

they are visited, and (iii) how long it takes to travel between them. Discrete-time random walks can capture the first two, but they lack a natural treatment of travel times between nodes. Continuous-time Markov chains (CTMCs) unify all three: their generator $Q$ specifies both transition probabilities and waiting times. This makes CTMCs a natural tool for modeling path distributions, and thus the ideal foundation for an embedding objective that preserves structure consistently across scales.

Building on these insights, we introduce Diffusion-Aligned Embeddings (DAE), a framework that aligns the path distributions of the high-dimensional and embedding-induced CTMCs. Our Path-KL objective provides formal guarantees of multiscale fidelity, admits an efficient GPU implementation, and consistently yields embeddings that preserve both local and global structure across a wide range of datasets. The remainder of the paper develops the theoretical framework (§2), presents the optimization algorithm (§3), and reports empirical evaluations (§4).

## 1.1 RELATED WORK

Most existing approaches to manifold learning are designed to favor either local or global structure. For example, t-SNE van der Maaten & Hinton (2008) and UMAP McInnes et al. (2018) excel at preserving local neighborhoods—ensuring that nearby points in high-dimensional space remain close in the embedding. However, these methods often distort global relationships, causing distant clusters to appear artificially close Kobak & Berens (2019). Conversely, classical techniques such as PCA Jolliffe (2002) and multidimensional scaling Borg & Groenen (2005) prioritize global structure by preserving variance or pairwise distances, yet frequently fail to capture the fine-grained local relationships that delineate cluster boundaries and smooth transitions Tenenbaum et al. (2000); Roweis & Saul (2000). More recent methods, including TriMAP Amid & Warmuth (2019) and PaCMAP Wang et al. (2021), attempt to balance local and global preservation, but rely on complex multi-term objectives and sensitive hyperparameters, and lack GPU implementations. Our work builds on these advances, seeking a unified framework that captures local and global structure while remaining amenable to large-scale datasets.

## 2 PATH DISTRIBUTIONS FOR LOW DIMENSIONAL EMBEDDINGS

Originally, diffusion models were introduced to mathematically characterize how heat propagates through a solid over time, tracing back to the seminal work of Baron de Fourier (1822).[2] The evolution of heat encodes information about the surface connectivity, curvature, and geometry. By analogy, the diffusion of probability on a data graph reveals fine-grained neighborhood structure at short timescales and global connectivity at longer ones.

### 2.1 FORMAL SETTING

To translate this geometric intuition of diffusion into a data-analytic setting, we represent a high-dimensional dataset as a weighted graph $G = (V, E, W)$, where nodes $V$ correspond to datapoints and edges $E$ encode neighborhood similarity. Diffusion processes on $G$ generalize the notion of heat flow: probability mass initiated at one node spreads to its neighbors at rates proportional to edge weights.

In continuous time, these dynamics are described by a generator matrix $Q \in \mathbb{R}^{n \times n}$. This matrix specifies both how quickly the process leaves each node and how likely it is to jump to each neighbor. Formally, $Q_{ij} \geq 0$ for $j \neq i$, and $Q_{ii} = -\sum_{j \neq i} Q_{ij}$ so that rows sum to zero. The negative diagonal entries $Q_{ii}$ encode the exit rate from node $i$, while the normalized off-diagonal entries $Q_{ij}/(-Q_{ii})$ give the probabilities of choosing the next neighbor.

From the generator $Q$, one can construct a family of transition matrices that describe the distribution of the diffusion at a future time $t$. Specifically, the dynamics are governed by the semi-group of diffusion operators

$$\mathcal{P}_t = e^{tQ}, \qquad t \geq 0, \tag{1}$$

---

[2]This line of research was later extended by Fick (1995), who formulated what is now known as Fick's law, and by Einstein (1905), who established connections to Brownian motion.

where $\mathcal{P}_t(i,j)$ is the probability of being at node $j$ after time $t$ when starting at $i$. The collection $\mathcal{P}_{t \geq 0}$ is called a semigroup of diffusion operators: it satisfies the composition rule

$$\mathcal{P}_s \mathcal{P}_t = \mathcal{P}_{s+t} \tag{2}$$

which formalizes the idea that running the diffusion for time $s$ and then $t$ is equivalent to running it for $s + t$ directly.

In this way, the generator $Q$ completely determines the stochastic dynamics of diffusion on the graph: it encodes both where the process moves next and how long it lingers at each node.

While $\mathcal{P}_t$ captures state probabilities at a fixed time $t$, it does not describe *how paths evolve across time*. To faithfully capture both local exploration and global mixing, we must consider the *path distribution* $(X_t)_{0 \leq t \leq T}$. It is described by a sequence

$$(i_0, \tau_0), ..., (i_m, \tau_m), \tag{3}$$

where $i_k$ is the visited state and $\tau_k \sim \exp(\lambda_{i_k})$ is the waiting time before moving to the next state, and the choice of next state is governed by conditional probabilities $p_{ij} = Q_{ij}/\lambda_i$.

Thus, a path encodes both (i) the visited states (via $p_{ij}$), and (ii) the length of a visit (via waiting times). Path distributions therefore provided richer descriptor of geometry compared to single-time marginals.

## 2.2 PATH-KL EMBEDDING

Intuitively, we would like the diffusion paths on the original high-dimensional graph and on the embedded graph to resemble one another. To formalize this, we require a way to compare two diffusion processes, specified by their respective generators. When comparing two diffusion processes with generators $Q$ and $\tilde{Q}$, a natural measure of discrepancy is the Kullback–Leibler (KL) divergence between their path distributions over a time horizon $T$:

$$\mathrm{KL}\left(\mathbb{P}_{Q,[0,T]} \mid \mathbb{P}_{\tilde{Q},[0,T]}\right) \coloneqq \mathbb{E}_{\mathbb{P}_{Q,[0,T]}}\left[\log \frac{d\mathbb{P}_{Q,[0,T]}}{d\mathbb{P}_{\tilde{Q},[0,T]}}\right], \tag{4}$$

where $\mathbb{P}_{Q,[0,T]}$ denotes the law of the trajectory $(X_t)_{0 \leq t \leq T}$ under generator $Q$.

However, this KL divergence grows linearly in $T$. We therefore consider the average divergence per unit time, i.e. the relative entropy rate:

$$\mathcal{E}(Q \parallel \tilde{Q}) \equiv \lim_{T \to \infty} \frac{1}{T} D_{\mathrm{KL}}(\mathbb{P}_{Q,[0,T]} \parallel \mathbb{P}_{\tilde{Q},[0,T]}). \tag{5}$$

Intuitively, $\mathbb{E}(Q \mid \tilde{Q})$ quantifies the long-run inefficiency (per unit time) of representing trajectories generated by $Q$ using the dynamics of $\tilde{Q}$.

**Decomposition of the Path-KL** The relative entropy rate between two continuous-time Markov chains admits a natural decomposition into interpretable terms. Let $\lambda_i = -Q_{ii}$ and $\tilde{\lambda}_i = -\tilde{Q}_{ii}$ denote the exit rates from state $i$, and $p_{ij} = Q_{ij}/\lambda_i$, $\tilde{p}_{ij} = \tilde{Q}_{ij}/\tilde{\lambda}_i$ the corresponding conditional transition probabilities. Then, we have the following decomposition.

**Lemma 2.1** (Path-KL Decomposition). *Let $Q$ and $\tilde{Q}$ be irreducible generator matrices on $n$ states with stationary distribution $\pi_Q$. Then the relative entropy rate is given by*

$$\mathcal{E}(Q \mid \tilde{Q}) = \sum_{i,j \neq i} \pi_{Q,i} Q_{ij} \log \frac{Q_{ij}}{\tilde{Q}_{ij}} + \sum_i \pi_{Q,i}(\lambda_{\tilde{Q},i} - \lambda_{Q,i}). \tag{6}$$

*Equivalently, it can be written as*

$$\mathcal{E}(Q \mid \tilde{Q}) = \sum_i \pi_{Q,i} \lambda_i \left[ \underbrace{\mathrm{KL}\big(p_i \mid \tilde{p}_i\big)}_{\textit{choice of next state}} + \underbrace{\mathrm{KL}\big(\exp(\lambda_i) \mid \exp(\tilde{\lambda}_i)\big)}_{\textit{waiting time}} \right], \tag{7}$$

*where $\lambda_i = -Q_{ii}$ and $\tilde{\lambda}_i = -\tilde{Q}_{ii}$ are exit rates, and $p_{ij} = Q_{ij}/\lambda_i$, $\tilde{p}_{ij} = \tilde{Q}_{ij}/\tilde{\lambda}_i$ are conditional transition probabilities.*

The proof is provided in Appendix A.

**Interpretation**  This decomposition highlights two complementary aspects of diffusion alignment. The term $\mathrm{KL}(p_i \mid \tilde{p}_i)$ measures discrepancies in the choice of the next state, reflecting how well local neighborhood transitions are preserved in the embedding. The term $\mathrm{KL}(\exp(\lambda_i) \mid \exp(\tilde{\lambda}_i))$ measures discrepancies in the waiting time distributions, reflecting how long the process lingers at each node. Since waiting times depend on node degrees, this term encodes both local density and, when aggregated over time, global mixing behavior.

Thus, *our goal* is to minimize the Path-KL divergence. By doing so, we will simultaneously align the direction of diffusion (which neighbors are chosen) and its how long the process dwells at each node, thereby obtaining embeddings that faithfully preserve structure across scales.

## 2.3 PATH-KL DIVERGENCE AS AN EMBEDDING OBJECTIVE

The decomposition in Lemma 2.1 provides a natural recipe for constructing an embedding objective. Given an embedding $\{y_i\}_{i=1}^n \subset \mathbb{R}^d$, we define a similarity kernel $K : \mathbb{R} \geq 0 \to \mathbb{R} > 0$ and specify the embedded generator by

$$\tilde{Q}_{ij}(y) = K\big(\|y_i - y_j\|^2\big), \quad i \neq j, \tag{8}$$

$$\tilde{Q}_{ii}(y) = -\sum_{j \neq i} \tilde{Q}_{ij}(y). \tag{9}$$

This construction guarantees that $\tilde{Q}(y)$ is a valid generator matrix, from which we obtain exit rates $\tilde{\lambda}_i(y)$ and conditional transition probabilities $\tilde{p}_{ij}(y)$ in the low-dimensional space.

Substituting $\tilde{Q}(y)$ into the decomposition of the relative entropy rate yields the Path-KL embedding objective:

$$\mathcal{L}(y) = \sum_i \pi_{Q,i} \lambda_i \Big[ \mathrm{KL}\left( p_i \,\|\, \tilde{p}_i(y) \right) + \mathrm{KL}\left( \exp(\lambda_i) \,\|\, \exp(\tilde{\lambda}_i(y)) \right) \Big]. \tag{10}$$

Minimizing this objective enforces the alignment of diffusion dynamics between the original high-dimensional graph $Q$ and the embedding-induced graph $\tilde{Q}(y)$. The first term penalizes discrepancies in the *choice of neighbors*, promoting preservation of local neighborhood structure. The second term penalizes discrepancies in the *waiting times*, encouraging embeddings that capture node degrees and global mixing behavior. Taken together, these two components ensure that the embedding preserves both fine-grained local geometry and large-scale global organization.

Moreover, unlike previous approaches, minimizing this objective provides formal guarantees on the fidelity with which the data structure is preserved, as we further discuss in the next section.

## 2.4 MULTISCALE PRESERVATION GUARANTEES

A notable property of the objective in Equation 10 is that minimizing it provides a formal guarantee of structural preservation. Moreover, it yields a practically verifiable criterion for assessing how well the low-dimensional embedding reflects the original data geometry. The central idea is the following: if the relative entropy rate between $Q$ and $\tilde{Q}$ is small, then the corresponding diffusion operators remain close across all timescales. We formalize this intuition in the following theorems.

**Theorem 2.2** (Generator Closeness). *Let $Q$ and $\tilde{Q}$ be irreducible generators with stationary distribution $\pi = \pi_Q$ and average exit rate $\bar{\lambda} = \sum_i \pi_i \lambda_i$. If $\mathcal{E}(Q \mid \tilde{Q}) \leq \epsilon$, then*

$$\|Q - \tilde{Q}\|_{1,\pi} \ \leq \ C_1 \sqrt{\bar{\lambda}\,\epsilon}, \tag{11}$$

*for an absolute constant $C_1$.*

This result shows that small Path-KL divergence guarantees the generators themselves are close. Since the generator encodes both the probabilities of transitioning to neighbors and the waiting times, this bound ensures that the embedding preserves the local rules governing the diffusion dynamics.

**Theorem 2.3** (Semi-group Closeness). *Under the same assumptions, the diffusion semi-groups* $\{P_t = e^{tQ}\}$ *and* $\{\tilde{P}_t = e^{t\tilde{Q}}\}$ *satisfy*

$$\|P_t - \tilde{P}_t\|_{\infty \to 1, \pi} \leq C_2 \sqrt{\bar{\lambda}} \epsilon \cdot \min(t, C_3), \tag{12}$$

*for all* $t \geq 0$*, where* $C_2, C_3$ *are absolute constants.*

The second theorem extends generator closeness to the full diffusion dynamics. At short times, the discrepancy between $P_t$ and $\tilde{P}_t$ grows at most linearly in $t$, ensuring faithful preservation of fine-scale neighborhoods. At long times, the discrepancy saturates, guaranteeing that global connectivity and mixing structure are retained.

Together, these bounds formalize the intuition that minimizing the Path-KL divergence enforces preservation of diffusion dynamics across all scales.

In contrast to heuristic objectives that balance local and global terms separately, our framework guarantees both simultaneously from a single probabilistic principle. This provides a unified foundation for dimensionality reduction: embeddings that minimize the Path-KL objective necessarily preserve both local geometry and global organization.

Proofs of both theorems are provided in Appendix B.

## 3    OPTIMIZATION ALGORITHM

Having established theoretical guarantees on the Path-KL divergence objective, we now turn to the practical challenge of minimizing equation 6 for large scale datasets. The generator $Q$ in equation 6 is generally a dense $n \times n$ matrix, requiring $O(n^2)$ storage and updates. To make optimization feasible, we instead restrict the class of admissible generators $Q$ constructed from sparse neighborhood graphs.

This design choice is inspired by results in manifold learning showing that $k$-nearest-neighbor graphs, when properly normalized, converge to the Laplace–Beltrami operator in the limit (Belkin & Niyogi, 2003; Hein et al., 2005; Coifman & Lafon, 2006; Von Luxburg et al., 2008). In other words, as the sample size increases, the discrete diffusion defined on the graph converges to the continuous diffusion on the underlying manifold, ensuring that local graph structure consistently reflects the manifold's geometry. While these results are asymptotic and rely on specific scaling conditions, they motivate relying on local neighborhoods to capture diffusion geometry. This sparse construction therefore reduces complexity to practically $O(nk)$ edges in most cases while retaining the essential structure of the data.

### 3.1    GRAPH CONSTRUCTION IN THE HIGH DIMENSIONAL SPACE

A naive algorithm optimizing the relative entropy rate in Equation 6 has a runtime of $O(|E|)$ per iteration. Therefore, the first step to to construct a sparse graph that adequately captures the local geometry of the data while keeping the number of edges $|E|$ manageable.

### 3.2    PARALLEL OPTIMIZATION

We cast the Path-KL objective into a per-pair decomposition that supports simple, parallel stochastic updates. This decomposition separates an attractive term $(-\log \tilde{Q}_{ij})$, which pulls neighboring points together, from a repulsive term $(+\tilde{Q}_{ij})$, which pushes points apart. This structure further enables an importance-weighted edge scheduler that prioritizes updates on the most influential pairs.

As we show in Theorem 3.1, the resulting estimator is unbiased and highly scalable. Concretely, the relative entropy rate we aim to minimize is proportional to

$$\mathcal{J}(Y) = \sum_{i \neq j} \pi_{i,Q} Q_{ij} \underbrace{-\log \tilde{Q}_{ij}(Y)}_{\text{attraction (pulls } i, j \text{ together)}} + \sum_{i \neq j} \pi_{i,Q} \underbrace{\alpha \tilde{Q}_{ij}(Y)}_{\text{repulsion (pushes } i, j \text{ apart)}}, \tag{13}$$

where each edge $(i, j)$ contributes an attraction term weighted by $\pi_{i,Q} Q_{ij}$ and a repulsion term weighted by $\pi_i \alpha$. This edge-wise decomposition suggests the following sampling scheme.

**Sampling Scheme.** Edges $(i, j)$ are sampled via importance sampling with probability $w_{ij} = \pi_{i,Q} Q_{ij}/P_{\max}$, where $P_{\max} := \max_{u,v} \pi_{Q,u} Q_{uv}$, and thus $(i, j)$ is sampled probability $w_{ij}$. When $(i, j)$ is drawn, we apply the attraction gradient $-\nabla \log \tilde{Q}_{ij}$. To obtain an unbiased estimator of the repulsion term for the node $i$,

$$\sum_{j \neq i} \pi_{i,Q} \, \alpha \, \tilde{Q}_{ij}(Y),$$

we additionally sample $n_{\mathrm{neg}}$ destination nodes $k_1, \ldots, k_{n_{\mathrm{neg}}}$ i.i.d uniformly from $\{1, \ldots, n\} \setminus \{i\}$.

For each sampled pair $(i, k_\ell)$ we apply the repulsion gradient $\nabla \tilde{Q}_{ik_\ell}(Y)$, multiplied by a per-source scaling factor, so that in expectation the total repulsive contribution equals the true term above. The full algorithm is summarized in Algorithm 1.

**Theorem 3.1** (Unbiased estimator). *Let $w_{ij} = \pi_{i,Q} Q_{ij}/P_{\max}$. Suppose each positive edge $(i, j)$ is sampled with frequency $w_{ij}$ and, when drawn, the attraction gradient $-\nabla \log \tilde{Q}_{ij}(Y)$ is applied. For each positive edge anchored at $i$, additionally draw $n_{neg}$ uniform destinations $j \neq i$ and apply the repulsion gradient $+\nabla \tilde{Q}_{ij}(Y)$, multiplied by*

$$\rho_i = \pi_{i,Q} \, \frac{\alpha}{P_{\max}} \, \frac{n-1}{n_{neg} \sum_{k:\,(i,k) \in E} w_{ik}}.$$

*Then, conditional on $Y$, the expected update satisfies*

$$\mathbb{E}[\Delta Y \mid Y] = \tfrac{1}{P_{\max}} \nabla \mathcal{J}(Y),$$

*i.e. the stochastic updates are unbiased for the full gradient of $\mathcal{J}$ up to the global constant $\frac{1}{P_{\max}}$ which can be absorbed into the learning rate.*

*Proof sketch.* The key insight is that the scaling factor $\rho_i$ exactly compensates for the non-uniform sampling frequencies. When positive edge $(i, j)$ is sampled with frequency $w_{ij} = \pi_{i,Q} Q_{ij}/P_{\max}$, the expected positive contribution becomes proportional to $\pi_{i,Q} Q_{ij} \nabla \log \tilde{Q}_{ij}(Y)$. Similarly, the careful choice of $\rho_i$ ensures that each negative pair $(i, j)$ contributes $\alpha \pi_{i,Q} \nabla \tilde{Q}_{ij}(Y)$ in expectation, despite being sampled uniformly rather than according to the true distribution. The factors involving $n_{\mathrm{neg}}$, $n-1$, and $\sum_k w_{ik}$ cancel exactly, yielding an unbiased estimator of the full gradient. The complete proof is given in Appendix C. $\qquad\square$

**Parallel Implementation.** In parallel SGD, when multiple workers attempt to update the same embedding vector $y_i$ simultaneously, **locking** mechanisms are typically used to ensure consistency—a worker must acquire an exclusive lock on $y_i$ before modifying it, forcing other workers to wait. This serialization creates computational bottlenecks, especially for high-degree nodes.

**Lock-Free Edge-SGD and Collision Rate** Because each stochastic update touches at most two node parameters—the endpoints of a sampled edge—workers can run without locks in the spirit of HOGWILD! (Recht et al., 2011). Let $q$ be a fixed distribution over edges $E$ from which each worker samples independently, and let $W$ denote the number of concurrent workers. We say two concurrent updates *collide* if they attempt to write the same node parameter $y_i$.

If we view the $W$ in-flight updates as $W$ i.i.d. edges $e_1, \ldots, e_W \sim q$, the expected number of colliding pairs in one update step is

$$\Lambda_W(q) = \binom{W}{2} \rho(q), \qquad \rho(q) = \Pr_{e,e' \sim q}[e \cap e' \neq \varnothing] = \sum_{v \in V} \left( \sum_{e \ni v} q(e) \right)^2 - \sum_{e \in E} q(e)^2.$$

Hence, by a union bound, the probability that *any* collision occurs in a step is at most $\binom{W}{2} \rho(q)$. Under uniform edge sampling ($q(e) = 1/m$ with $m = |E|$) this simplifies to

$$\rho(q) = \frac{\sum_{v \in V} d_v^2 - m}{m^2},$$

**Algorithm 1:** Diffusion-aligned embedding with unbiased row-normalized negatives

**Data:** Dataset $X$, parameters $k, K, \alpha, d, T, B, n_{\text{neg}}$
**Result:** Embedding $Y$
**Graph:** $(E, Q, \pi) \leftarrow \texttt{build\_graph}(X, k, K)$;
**Weights:** $w_{ij} \leftarrow \dfrac{\pi_{i,Q} Q_{ij}}{P_{\max}}$ where $P_{\max} = \max_{(u,v) \in E} \pi_{u,Q} Q_{uv}$;
**Repulsion scales:** $\rho_i \leftarrow \pi_i \dfrac{\alpha}{P_{\max}} \dfrac{n-1}{n_{\text{neg}} \sum_k w_{ik}}$;
**Init:** $Y \leftarrow \texttt{init\_embedding}(X, E, Q, d)$;
**for** $t = 1$ **to** $T$ **do**
   $\sigma \leftarrow \texttt{stride\_permute}(E)$;   $\{B_b\} \leftarrow \texttt{partition\_blocks}(\sigma, B)$;
   // To maximize cache hits
   **foreach** $b$ *in parallel* **do**
      **foreach** $(i, j) \in B_b$ **do**
         **if** $\texttt{should\_fire}(w_{ij}, t)$ **then**
            $s \leftarrow s(y_i, y_j)$
            $g \leftarrow \texttt{attraction\_grad}(s, Q_{ij})$;
            **atomic** $y_i \leftarrow y_i - \eta_t\, g$; ;
              **atomic** $y_j \leftarrow y_j + \eta_t\, g$;
            **for** $m = 1$ **to** $n_{neg}$ **do**
              $j^- \leftarrow \texttt{sample\_uniform\_dest}(n, \text{exclude } i)$
              $s^- \leftarrow s(y_i, y_{j^-}); r \leftarrow \texttt{repulsion\_grad}(s^-)$
              **atomic** $y_i \leftarrow y_i - \eta_t\, \rho_i\, r$;
                **atomic** $y_{j^-} \leftarrow y_{j^-} + \eta_t\, \rho_i\, r$;

**return** $Y$;

so for sparse graphs with bounded degree second moment (i.e., $\sum_v d_v^2 = \Theta(n)$ and $m = \Theta(n)$) we obtain $\rho(q) = \Theta(1/n)$ and thus

$$\Pr(\text{collision in a step}) \;=\; O\!\left(\frac{W^2}{n}\right).$$

Consequently, as long as $W \ll \sqrt{n}$, collisions are rare and the bias introduced by stale or lost writes is negligible (cf. Recht et al., 2011).[3]

On CPU, where the number of workers $W$ is modest, this Hogwild assumption is reasonable and provides excellent speedups without synchronization overhead. On GPU, however, thousands of threads update concurrently and the $O(W^2/n)$ collision probability is no longer negligible. In that setting our implementation employs hardware-supported `atomicAdd` operations to accumulate increments safely, ensuring correctness while still exploiting massive parallelism.

## 4 EXPERIMENTS

Having established that minimizing the Path-KL objective provides formal guarantees on multiscale structure preservation, we now investigate whether these theoretical advantages translate to practical improvements. We ask: (1) Does jointly optimizing transition probabilities and waiting times outperform methods focused on pairwise similarities? (2) How robust is our approach across biological contexts and neighborhood scales? (3) Can Path-KL effectively balance local and global structure?

To comprehensively answer these questions, we benchmark DAE against popular embedding methods including UMAP (McInnes et al., 2018), t-SNE (van der Maaten & Hinton, 2008),

---

[3]In graphs with hubs (e.g., stars), $\sum_v d_v^2$ can be large and $\rho(q)$ need not vanish under uniform sampling; our expression makes this dependence explicit and also applies to non-uniform $q$. See Appendix A for a proof and for scheduling variants.

PaCMAP (Wang et al., 2021), PHATE (Moon et al., 2019), and TriMAP (Amid & Warmuth, 2019). We selected five diverse single-cell RNA-seq datasets representing key biological use-cases: (1) simulated trajectory data with known ground-truth branching structure (chronocellsim) (Fang et al., 2025); (2) perturbation responses in peripheral blood mononuclear cells (PBMC) treated with interferon-$\beta$, capturing subtle biological variations (kang dataset) (Kobak & Berens, 2019); (3) monocyte response to drug perturbations (monocytedrug), reflecting challenging pharmacologically induced changes (Resztak et al., 2023); (4) a canonical reference PBMC dataset widely used for benchmarking local cell-type structure (pbmc3k) (Chari & Gorin, 2023); and (5) developmental single-cell data (vu) capturing clear global differentiation trajectories (Vu et al., 2022).

To assess robustness, we systematically vary the neighborhood parameter $k \in \{15, 25, 30, 40, 50, 60, 75\}$ and assess 18 metrics capturing local and global structure via the ZADU framework (Jeon et al., 2023). Among these, we focus on three complementary metrics: trustworthiness quantifies local neighborhood preservation (whether embedding neighbors were truly close in the original space), while steadiness and cohesiveness measure inter-cluster reliability—steadiness detects false groups (distinct clusters artificially merged) and cohesiveness identifies missing groups (true clusters becoming fragmented). Together, these metrics assess whether embeddings preserve both fine-grained cell relationships and population-level organization critical for biological interpretation.

**Experimental Design**   We harmonize hyperparameters so that each method operates on comparable neighborhoods: UMAP (`n_neighbors = k`), t-SNE (`perplexity = k/3`), TriMAP (`n_inliers = k/5` and `n_outliers = k/10`), and DAE using the same k-NN graph. While our implementation largely follows the default settings of each method, we ensure that neighborhood sizes remain consistent across techniques to allow for a fair comparison. We show comprehensive results across all datasets and metrics in Figure 1, with additional qualitative and quantitative experiments on synthetic datasets and detailed embeddings provided in the Appendix. We train all methods for the same number of iterations, using spectral initialization and no tuning. U less otherwise specified we use a student$-t$ kernel function.

**Scalability of CUDA implementation**   We verified that our CUDA implementation can scale to large datasets. We filtered the CELLxGENE single-cell RNA-seq census to human, healthy samples generated with the 10x Genomics $3'$ v3 assay (Program et al., 2025). We then grouped cells by tissue and selected the largest dataset, which corresponded to brain. After standard preprocessing (see Appendix D for details), we retained $\tilde{9}.5$M cells and 5,000 genes. We corrected for batch effects (`dataset_id` and `donor_id`) using scVI (Lopez et al., 2018) with 50 latent dimensions. We then computed both UMAP (see cuML, Raschka et al., 2020) and DAE embeddings on scVI's latent space. Runtimes were comparable: UMAP ran in 65s, DAE in 225s on an NVIDIA H100 GPU. Both Figure 9 plots the DAE embeddings, while UMAP embeddings are shown in Appendix D.

## 5   CONCLUSION

This paper introduced DAE, which formulates dimensionality reduction as aligning diffusion processes between high- and low-dimensional spaces. By minimizing the Path-KL divergence—which uniquely captures both transition probabilities and waiting times of continuous-time random walks—we proved formal bounds on generator and semigroup closeness, guaranteeing structure preservation across scales. Our optimization algorithm decomposes this objective into attraction-repulsion terms with an unbiased gradient estimator, enabling efficient parallel implementation. Experiments on single-cell RNA-seq datasets showed DAE consistently preserves both local neighborhoods and global structure, while our CUDA implementation scales to millions of cells with competitive runtime. The Path-KL framework provides theoretical guarantees that complement existing diffusion-based methods. The code for DAE would be released publicly available with CPU and GPU.

Figure 1: Relative embedding quality across methods and datasets. We evaluate DAE against UMAP, t-SNE, PaCMAP, PHATE, and TriMAP using three complementary metrics: **Trustworthiness** (local neighborhood preservation), **Steadiness** (avoiding false groups where distinct clusters merge), and **Cohesiveness** (avoiding missing groups where true clusters fragment). Results shown across varying neighborhood sizes ($k \in \{15, 25, 30, 40, 50, 60, 75\}$) demonstrate that DAE consistently achieves strong performance in preserving both local structure and inter-cluster relationships, validating that our Path-KL objective's theoretical guarantees translate to practical improvements in biological embeddings.

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

# A    PROOF OF LEMMA 2.1

$$\lim_{T \to \infty} \frac{1}{T} \text{KL}(\mathbb{P}_{Q,[0,T]} || \mathbb{P}_{\tilde{Q},[0,T]}) = \sum_{j \neq i} \pi_{Q,i} Q_{ij} \log \frac{Q_{ij}}{\tilde{Q}_{ij}} + \sum_{i} \pi_{Q,i}(\lambda_{\tilde{Q},i} - \lambda_{Q,i}) \tag{14}$$

$$= \sum_{j \neq i} \pi_{Q,i} \lambda_{Q,i} p_{ij} \log \frac{\lambda_{Q,i} p_{Q,ij}}{\lambda_{\tilde{Q},i} p_{\tilde{Q},ij}} + \sum_{i} \pi_{Q,i}(\lambda_{\tilde{Q},i} - \lambda_{Q,i}) \tag{15}$$

$$= \sum_{j \neq i} \pi_{Q,i} \lambda_{Q,i} p_{Q,ij} \log \frac{p_{Q,ij}}{p_{\tilde{Q},ij}} + \sum_{i} \pi_{Q,i} \left( \lambda_{Q,i} \log \frac{\lambda_{Q,i}}{\lambda_{\tilde{Q},i}} - (\lambda_{\tilde{Q},i} - \lambda_{Q,i}) \right) \tag{16}$$

$$= \sum_{i} \pi_{Q,i} \lambda_{Q,i} \underbrace{\sum_{j \neq i} p_{Q,ij} \log \frac{p_{Q,ij}}{p_{\tilde{Q},ij}}}_{\text{KL}(p_{Q,i} || p_{\tilde{Q},i})} + \sum_{i} \pi_{Q,i} \lambda_{Q,i} \underbrace{\left( \log \frac{\lambda_{Q,i}}{\lambda_{\tilde{Q},i}} - \frac{(\lambda_{\tilde{Q},i} - \lambda_{Q,i})}{\lambda_{Q,i}} \right)}_{\text{KL}(\text{Exp}(\lambda_{Q,i}) || \text{Exp}(\lambda_{\tilde{Q},i}))} \tag{17}$$

$$= \sum_{i} \pi_{Q,i} \lambda_{Q,i} \left( \underbrace{\text{KL}(p_{Q,i} || p_{\tilde{Q},i})}_{\text{Choice of next state}} + \underbrace{\text{KL}(\text{Exp}(\lambda_{Q,i} || \lambda_{\tilde{Q},i})}_{\text{waiting times}} \right) \tag{18}$$

# B    FROM PATHWISE KL TO GENERATOR AND SEMIGROUP CLOSENESS

Let $\mathcal{S}$ be a finite state space. Let $Q$ and $\tilde{Q}$ be CTMC generators on $\mathcal{S}$ with jump rates $\lambda_i = -Q_{ii} > 0$ and post-jump kernels $p_{ij} = Q_{ij}/\lambda_i$ for $j \neq i$ (and likewise $\tilde{\lambda}_i, \tilde{p}_i$ for $\tilde{Q}$). Assume $Q$ is irreducible and let $\pi = \pi_Q$ be its stationary distribution. Define

$$\bar{\lambda} = \sum_{i} \pi_i \lambda_i, \qquad \psi(x) = x - 1 - \log x, \quad x > 0.$$

Assume support inclusion so that $\epsilon < \infty$:

$$\epsilon \equiv \lim_{T \to \infty} \frac{1}{T} D_{\text{KL}}(\mathbb{P}_{Q,[0,T]} || \mathbb{P}_{\tilde{Q},[0,T]}) = \sum_{i} \pi_i \lambda_i \left[ \psi\left(\frac{\tilde{\lambda}_i}{\lambda_i}\right) + D_{\text{KL}}(p_i || \tilde{p}_i) \right]. \tag{A.1}$$

**Lemma B.1** (A sharp elementary inequality)**.** *For all $x > 0$,*

$$(x - 1)^2 \leq 2(x + 1)\psi(x), \qquad \psi(x) = x - 1 - \log x.$$

*Proof.* Let $g(x) = 2(x+1)\psi(x) - (x-1)^2$. Then $g(1) = g'(1) = 0$ and $g''(x) = 2\left(1 - \frac{1}{x} + \frac{1}{x^2}\right) > 0$ for all $x > 0$. Hence $g(x) \geq 0$.  $\square$

**Theorem B.2** (Rate closeness)**.** *Let $S = \sum_{i} \pi_i |\lambda_i - \tilde{\lambda}_i|$. Then*

$$S \leq \epsilon + \sqrt{\epsilon^2 + 4\epsilon\bar{\lambda}} \leq 2\epsilon + 2\sqrt{\epsilon\bar{\lambda}} \leq 4\sqrt{\epsilon\bar{\lambda}} \quad \text{if } \epsilon \leq \bar{\lambda}.$$

*Proof.* By Lemma B.1 with $x = \tilde{\lambda}_i/\lambda_i$,

$$|\lambda_i - \tilde{\lambda}_i| \leq \lambda_i \sqrt{2\left(1 + \frac{\tilde{\lambda}_i}{\lambda_i}\right)\psi\left(\frac{\tilde{\lambda}_i}{\lambda_i}\right)}.$$

Summing with weights $\pi_i$ and applying Cauchy–Schwarz,

$$S \leq \sqrt{2}\left(\sum_{i} \pi_i \lambda_i \psi\left(\frac{\tilde{\lambda}_i}{\lambda_i}\right)\right)^{1/2}\left(\sum_{i} \pi_i(\lambda_i + \tilde{\lambda}_i)\right)^{1/2}.$$

By equation A.1, $\sum_{i} \pi_i \lambda_i \psi(\tilde{\lambda}_i/\lambda_i) \leq \epsilon$. Also, $\sum_{i} \pi_i(\lambda_i + \tilde{\lambda}_i) \leq 2\bar{\lambda} + S$. Hence $S \leq \sqrt{2\epsilon}\sqrt{2\bar{\lambda} + S}$, which is equivalent to $S^2 - 2\epsilon S - 4\epsilon\bar{\lambda} \leq 0$. The claimed bounds follow.  $\square$

**Lemma B.3** (Pinsker and the jump kernels)**.** *Let $T = \sum_i \pi_i \lambda_i TV(p_i, \tilde{p}_i)$, where $TV(\cdot, \cdot) = \frac{1}{2}\|\cdot - \cdot\|_1$. Then*

$$T \le \frac{1}{\sqrt{2}} \sqrt{\bar{\lambda}} \, \epsilon.$$

*Proof.* Pinsker gives $TV(p_i, \tilde{p}_i) \le \sqrt{D_{\mathrm{KL}}(p_i \| \tilde{p}_i)/2}$. By Cauchy–Schwarz,

$$T \le \sum_i \pi_i \lambda_i \sqrt{\tfrac{1}{2} D_{\mathrm{KL}}(p_i \| \tilde{p}_i)} \le \frac{1}{\sqrt{2}} \sqrt{\Big( \sum_i \pi_i \lambda_i \Big) \Big( \sum_i \pi_i \lambda_i D_{\mathrm{KL}}(p_i \| \tilde{p}_i) \Big)},$$

and the second factor is $\le \epsilon$ by equation A.1. $\qquad\square$

**Definition B.4** (Weighted operator norm)**.** *For a linear operator $G$ on functions $f : \mathcal{S} \to \mathbb{R}$ set*

$$\|G\|_{\infty \to 1, \pi} = \sup_{\|f\|_\infty \le 1} \sum_i \pi_i |(Gf)_i|.$$

**Lemma B.5** (Mixing of the skeleton and $d_Q(s)$)**.** *Fix $\alpha \ge \max_i \lambda_i$ and set $R = I + \alpha^{-1} Q$. If $R$ is irreducible and aperiodic, there exists $\rho \in [0, 1)$ with*

$$\sup_{f \perp 1} \frac{\|R^k f\|_{2, \pi}}{\|f\|_{2, \pi}} \le \rho^k.$$

*Consequently, with $\lambda = \alpha(1 - \rho)$,*

$$d_Q(s) := \sup_i TV\big(P_s(i, \cdot), \pi\big) \le \tfrac{1}{2} \sqrt{\tfrac{1}{\pi_{\min}} - 1} \, e^{-\lambda s}, \qquad s \ge 0.$$

*Proof.* Standard Perron–Frobenius/spectral arguments for finite chains imply $\rho < 1$. Using $\|f\|_{2,\pi}^2 - \|Rf\|_{2,\pi}^2 = \sum_j \pi_j \mathrm{Var}_{R(j, \cdot)}(f) \ge 0$, equality forces $f$ to be constant on the strongly connected graph, hence a scalar multiple of 1. The TV bound follows from Jensen and the $L^2(\pi)$ contraction together with $P_s = \mathbb{E}[R^{N_s}]$ with $N_s \sim \mathrm{Poisson}(\alpha s)$. $\qquad\square$

**Proposition B.6** (Generator closeness)**.** *With $S, T$ as above,*

$$\|Q - \tilde{Q}\|_{\infty \to 1, \pi} \le 2S + 2T.$$

*Consequently, if $\epsilon \le \bar{\lambda}$ then*

$$\|Q - \tilde{Q}\|_{\infty \to 1, \pi} \le (8 + \sqrt{2}) \sqrt{\bar{\lambda}} \, \epsilon.$$

*Proof.* For $f$ with $\|f\|_\infty \le 1$,

$$\sum_i \pi_i |(Q - \tilde{Q})f(i)| \le \sum_i \pi_i |\lambda_i - \tilde{\lambda}_i| \, |f(i)| + \sum_{i \ne j} \pi_i |\lambda_i p_{ij} - \tilde{\lambda}_i \tilde{p}_{ij}| \, |f(j)|$$

$$\le \sum_i \pi_i |\lambda_i - \tilde{\lambda}_i| + \sum_{i \ne j} \pi_i \lambda_i |p_{ij} - \tilde{p}_{ij}| + \sum_{i \ne j} \pi_i \tilde{p}_{ij} |\lambda_i - \tilde{\lambda}_i|$$

$$\le 2 \sum_i \pi_i |\lambda_i - \tilde{\lambda}_i| + 2 \sum_i \pi_i \lambda_i TV(p_i, \tilde{p}_i).$$

Take the supremum over $\|f\|_\infty \le 1$. Combine with Theorems B.2 and Lemma B.3. $\qquad\square$

**Theorem B.7** (Semigroup closeness)**.** *For all $t \ge 0$,*

$$\|P_t - \tilde{P}_t\|_{\infty \to 1, \pi} \le 2 \|Q - \tilde{Q}\|_{\infty \to 1, \pi} \int_0^t d_{\tilde{Q}}(s) \, ds \le \|Q - \tilde{Q}\|_{\infty \to 1, \pi} \min\left\{ t, \frac{1}{\lambda} \sqrt{\frac{1}{\pi_{\min}} - 1} \right\}.$$

*In particular, if $\epsilon \le \bar{\lambda}$ then*

$$\|P_t - \tilde{P}_t\|_{\infty \to 1, \pi} \le (8 + \sqrt{2}) \sqrt{\bar{\lambda}} \, \epsilon \cdot \min\left\{ t, \frac{1}{\lambda} \sqrt{\frac{1}{\pi_{\min}} - 1} \right\}.$$

*Proof.* Duhamel's formula gives $P_t - \tilde{P}_t = \int_0^t P_{t-s}(Q - \tilde{Q})(\tilde{P}_s - \tilde{\Pi}) \, ds$, since $(Q - \tilde{Q})\tilde{\Pi} = 0$. Using $\|P_{t-s}\|_{1 \to 1, \pi} \le 1$, $\|\tilde{P}_s - \tilde{\Pi}\|_{\infty \to \infty} = 2 \, d_{\tilde{Q}}(s)$, and Proposition B.6 yields the first inequality. The second follows from Lemma B.5 and the trivial bound $\int_0^t d_{\tilde{Q}}(s) \, ds \le t$. $\qquad\square$

## C    PROOF OF THEOREM 3.1

*Proof.* Let $(G, E, Q)$ be the high-dimensional data graph. If we sample a positive edge $(i, j) \in E$ with frequency $w_{ij}$ and apply the attraction gradient $-\nabla \log \tilde{Q}_{ij}(Y)$ to the embedding, then in expectation the positive contribution is

$$\mathbb{E}[\Delta Y_{(i,j),\text{pos}}] = w_{ij} \left( - \nabla \log \tilde{Q}_{ij}(Y) \right)$$
$$= \frac{1}{P_{\max}} \pi_{i,Q} Q_{ij} \left( - \nabla \log \tilde{Q}_{ij}(Y) \right),$$

since $w_{ij} = \pi_i Q_{ij}/P_{\max}$.

For the repulsive component, fix a source node $i \in [n]$. Each positive update anchored at $i$ triggers $n_{\text{neg}}$ uniform draws $j \in \{1, \dots, n\} \setminus \{i\}$. Thus a particular destination $j$ is selected with frequency

$$\frac{n_{\text{neg}}}{n-1} \sum_{k:\, (i,k)\in E} w_{ik}.$$

Multiplying the repulsion gradient $\nabla \tilde{Q}_{ij}(Y)$ by the scaling factor

$$\rho_i = \pi_{i,Q} \frac{\alpha}{P_{\max}} \frac{n-1}{n_{\text{neg}} \sum_{k:\, (i,k)\in E} w_{ik}}$$

yields

$$\mathbb{E}[\Delta Y_{(i,j),\text{neg}}] = \rho_i \cdot \left( \tfrac{n_{\text{neg}}}{n-1} \sum_{k:\, (i,k)\in E} w_{ik} \right) \cdot \nabla \tilde{Q}_{ij}(Y)$$
$$= \frac{\alpha}{P_{\max}} \pi_{i,Q} \cdot \nabla \tilde{Q}_{ij}(Y),$$

where the factors $(n-1)$, $n_{\text{neg}}$, and $\sum_{k:\, (i,k)\in E} w_{ik}$ cancel exactly.

Summing the terms over all $i$ and $j \neq i$ gives

$$\mathbb{E}[\Delta Y \mid Y] = \sum_{i,j\neq i} \left( \mathbb{E}[\Delta Y_{(i,j),\text{pos}}] + \mathbb{E}[\Delta Y_{(i,j),\text{neg}}] \right) \tag{19}$$

$$= - \sum_{i,j\neq i} \frac{1}{P_{\max}} \pi_{i,Q} Q_{ij} \left( \nabla \log \tilde{Q}_{ij}(Y) \right) + \sum_{i,j\neq i} \frac{\alpha}{P_{\max}} \pi_{i,Q} \cdot \nabla \tilde{Q}_{ij}(Y) \tag{20}$$

$$= \frac{1}{P_{\max}} \left( - \sum_{(i,j)\in E} \pi_{i,Q} Q_{ij} \nabla \log \tilde{Q}_{ij} + \sum_{i,j\neq i} \alpha\, \pi_{i,Q}\, \nabla \tilde{Q}_{ij}(Y) \right) \tag{21}$$

$$= \frac{1}{P_{\max}} \nabla \mathcal{J}(Y). \tag{22}$$

$\square$

## D    ADDITIONAL EXPERIMENTAL DETAILS

## D.1 Additional Qualitative and Benchmark Experiments on Toy Datasets

Here we provide qualitative visualizations on standard synthetic manifold learning benchmarks with known ground-truth structure. These toy datasets allow us to directly assess each method's ability to preserve topology, handle non-convex geometries, and unfold complex manifolds where the true structure is known a priori.

Figure 2: Comprehensive 2D embedding comparisons across methods. We visualize embeddings produced by DAE/, UMAP, t-SNE, and TriMAP on each dataset, demonstrating qualitative differences in structure preservation. DAE consistently maintains both local neighborhood relationships and global topology across diverse biological contexts.

## D.2 Scaling-laws and runtime

We analyze the computational efficiency and scaling behavior of DAE compared to existing methods. The computational time increases and is extremely close linear runtime as seen in figure **??**

Table 1: Mean and standard deviation for ranks across local and global embedding quality metrics for toy datasets. Rankings for each metric are averaged over 5 repetitions with different random seeds. Bold denotes best performance. DAE performs comparatively well, ranking best or second best across 5 out of 6 datasets.

|          | Swiss Roll (w/ Hole) | Moons | Digits |
|----------|:---:|:---:|:---:|
| t-SNE    | $3.3 \pm 1.81$ | $\mathbf{3.0 \pm 2.08}$ | $\mathbf{2.8 \pm 1.03}$ |
| UMAP     | $3.6 \pm 0.87$ | $3.4 \pm 1.49$ | $5.4 \pm 1.7$ |
| TriMap   | $3.9 \pm 2.29$ | $4.1 \pm 1.52$ | $4.4 \pm 1.57$ |
| PaCMAP   | $3.5 \pm 1.55$ | $\mathbf{3.0 \pm 1.24}$ | $5.7 \pm 1.24$ |
| LocalMAP | $3.9 \pm 1.36$ | $3.5 \pm 0.45$ | $4.7 \pm 1.64$ |
| DAE      | $\mathbf{2.8 \pm 1.59}$ | $4.0 \pm 2.48$ | $3.6 \pm 1.17$ |

|          | Swiss Roll | Circles | S - Curve |
|----------|:---:|:---:|:---:|
| t-SNE    | $3.3 \pm 1.81$ | $4.4 \pm 1.81$ | $4.3 \pm 1.63$ |
| UMAP     | $3.6 \pm 0.87$ | $4.2 \pm 1.49$ | $3.9 \pm 1.7$ |
| TriMap   | $3.9 \pm 2.29$ | $4.0 \pm 1.45$ | $3.4 \pm 2.2$ |
| PaCMAP   | $3.5 \pm 1.55$ | $\mathbf{2.2 \pm 1.19}$ | $3.1 \pm 1.1$ |
| LocalMAP | $3.9 \pm 1.36$ | $3.7 \pm 1.15$ | $4.0 \pm 1.33$ |
| DAE      | $\mathbf{2.8 \pm 1.59}$ | $2.5 \pm 1.72$ | $\mathbf{2.3 \pm 1.14}$ |

Figure 3: Quantitative performance metrics across all 18 ZADU evaluation criteria. Extended results showing mean and standard deviation across five random seeds for each method-dataset combination. Bold values indicate best performance, with DAE achieving top scores in metrics that balance local and global structure preservation.

### D.3 CUDA LARGE-SCALE EXPERIMENT

We constructed the benchmarking dataset by filtering the CELLxGENE corpus with the following parameters: `species = Homo sapiens`, `disease = normal`, and `assay = 10x 3' v3`. After filtering, datasets were grouped by `tissue_general` and ranked by cell count. We then selected the largest dataset, corresponding to brain. Cells were further filtered to retain only those with `n_raw > 300`. We selected 5,000 highly variable genes.

We trained a scVI model with the following settings. Model initialization used `n_hidden=512`, `n_latent=50`, `n_layers=2` and `gene_likelihood="nb"`. Training was performed with `max_epochs=100`, `train_size=0.9`, and `batch_size=50,000`. The optimizer plan was set with `lr=1e-4` and `n_epochs_kl_warmup=20`.

All training and embedding runs used a compute node equipped with an NVIDIA H100 GPU, 100 GB of RAM, and 24 CPU cores (CUDA 12.2).

Figure 4: Synthetic manifold learning benchmark using Swiss roll with hole dataset. This challenging topology tests each method's ability to (a) preserve the continuous manifold structure, (b) maintain the hole (avoiding spurious connections), and (c) correctly unfold the spiral. DAE's Path-KL objective naturally respects the manifold's geometry by modeling diffusion paths rather than forcing Euclidean distances. The ablation shows how different embedding change just by virtue of increasing/decreasing the number of dataspoints.

Figure 7: Synthetic manifold learning benchmark using Swiss roll with hole dataset. This challenging topology tests each method's ability to (a) preserve the continuous manifold structure, (b) maintain the hole (avoiding spurious connections), and (c) correctly unfold the spiral. DAE's Path-KL objective naturally respects the manifold's geometry by modeling diffusion paths rather than forcing Euclidean distances. The ablation shows how different embedding change just by virtue of increasing/decreasing the number of dataspoints.

Figure 8: UMAP embeddings of 9.5M brain datasets from the CELLxGENE corpus.

Figure 9: DAE embeddings of 9.5M brain cells from healthy human tissue (CELLxGENE). Our CUDA implementation scales efficiently to datasets of this size.

