# OpenReview forum: "Diffusion Aligned Embeddings"
_ICLR.cc/2026/Conference — ICLR 2026 Conference Withdrawn Submission_

### Official Review · Reviewer_LGBa · 2025-10-28

**Soundness:** 3
**Presentation:** 2
**Contribution:** 2
**Rating:** 2
**Confidence:** 2

**Summary:**

The paper introduces DAE (Diffusion-Aligned Embeddings), a dimensionality-reduction framework that aligns continuous-time diffusion CTMC path distributions between the high-dimensional data graph and a learned low-dimensional graph by minimizing a Path-KL relative entropy rate between path laws. The authors prove two main theoretical guarantees: generator closeness (small Path-KL implies the two generators are close in a weighted operator norm) and semigroup closeness (closeness of the diffusion operators across timescales). They also provide a computationally efficient algorithm for this objective. Empirically, DAE is compared to UMAP, t-SNE, PaCMAP, PHATE and TriMAP on several single-cell RNA-seq datasets using the ZADU framework and the results show consistent improvement in preserving both local and global structure.

**Strengths:**

Formulating embedding as alignment of CTMC path distributions is a good idea that unifies local and global preservation through a single probabilistic objective (Path-KL), rather than ad-hoc balancing terms.

The paper’s theoretical development is one of its strongest aspects. The paper contains nontrivial, technically meaningful theorems linking Path-KL to generator and semigroup closeness.

The experiments on multiple realistic single-cell RNA-seq datasets are appropriate for the paper’s target application and show measurable gains.

**Weaknesses:**

The theoretical results assume an irreducible generator and, implicitly, accurate knowledge of high-dimensional generator Q and stationary π. In practice, the method builds a sparse k-NN graph; the paper does not fully quantify how approximations introduced by sparse graph construction affect the Path-KL bounds. The gap between theory (dense/ideal generator) and sparse practical graph is not fully characterized.

Key algorithmic parameters (kernel K and its bandwidth, α controlling repulsion weight, nneg, importance sampling schedule, Pmax normalization) are only lightly discussed. Ablations showing sensitivity and guidance for practitioners are missing.


All experiments are on single-cell RNA-seq datasets. While that’s an important and challenging domain, it’s quite specific. The authors claim DAE is a general embedding method for any high-dimensional data, but there’s no evidence it works outside biology. Comparisons to broader synthetic manifolds or vision/text representations (where ground truth geometry is known) would strengthen generality claims.


The paper compares against UMAP, t-SNE, PHATE, PaCMAP, and TriMAP, but it explicitly says it used default parameters.
This reduces my confidence in DAE as these methods are very sensitive to parameters like perplexity or neighbor count. DAE might look better simply because the baselines weren’t tuned

On page 15, a figure appears to be missing. In addition, the final figures are not discussed in the text, so their purpose and key message are unclear. A discussion interpreting these figures should be added to help readers understand their relevance.
A minor point: the last two figures significantly increase the PDF file size, making it difficult to open. Consider optimizing or resizing them to improve accessibility.

**Questions:**

1) Theorems 2.2–2.3 assume an irreducible generator Q with stationary distribution π, but the implementation uses a sparse k-NN graph. How sensitive are the Path-KL bounds to this sparsification? Could you provide empirical evidence or an analytical bound on the error introduced by using sparse Q?

2) Have you conducted parameter sweeps over α, nneg, or kernel bandwidth? If so, can you share sensitivity plots or ranges where performance remains stable? Could you provide practical guidance for choosing Pmax and the importance-sampling schedule across different dataset scales?

3) Have you evaluated DAE on synthetic manifolds (e.g., Swiss roll, S-curve) or vision/text embeddings to confirm its generality? Can you discuss more the Figures in the appendix?

4) Why were the baselines run with default hyperparameters instead of tuned settings per dataset? Could you provide a supplementary tuning experiment (e.g., varying n_neighbors or perplexity) to show that DAE’s improvement persists under optimized baselines?

---

### Official Review · Reviewer_EzWE · 2025-10-28

**Soundness:** 3
**Presentation:** 2
**Contribution:** 2
**Rating:** 2
**Confidence:** 5

**Summary:**

This paper proposes to tackle the problem of estimating low-dimensional embeddings by aligning a diffusion process defined on the graph spanned by the data points in low-dimensional (target) space with the diffusion process on the graph spanned by the same points in the (given) high-dimensional space. They suggest to optimize this alignment through Path-KL divergence and provide theoretical guarantees for this optimization.

**Strengths:**

-	The theoretical idea of optimizing through Path-KL seems interesting and novel, albeit the practical implementation of it seems unclear (see Weaknesses).

**Weaknesses:**

-	The experimental part is sloppy at best and **lacks mentioning or discussion of the actually achieved results** – what is the performance in terms of reconstruction quality comparing DAE against existing methods, what can we draw from that?
- The **reporting of experimental results on their method is missing**, DAE is not to be seen in Figure 1. **There are no further figures or tables in the main paper**. Also in the Appendix, there are **no reported metrics on the real-world datasets**.
-	Even for competitors, it is unclear what the achieved results show – the metrics were averaged across datasets, which seems odd. In the Appendix, they instead provide mean and std across different metrics, which also does not make sense. The achieved **results per metric and per dataset** should be shown for each method, the standard benchmark metrics from the field should be ideally used for this.
-	The work misses a proper discussion and comparison to relevant related work, in particular works that are targeted to (very) large-scale datasets and efficiency [1-3], as well as variants that improve the reconstruction quality compared to existing works, e.g.  [4], or explicitly reconstruct local and global properties at once [5]. [1-3] should scale (more) effectively to the 5 million point dataset. [4] sets a SOTA performance for reconstruction quality.

[1] Tang, J et al. *Visualizing Large-scale and High-dimensional Data.* WWW 2016.

[2] Artemenkov, A, Panov, M. *NCvis: Noise contrastive approach for scalable visualization.* WWW 2020.

[3] Linderman, G et al. *Fast interpolation-based t-SNE for improved visualization of single-cell RNA-seq data.* Nature Methods 2019.

[4] Narayan, A, et al. *Assessing single-cell transcriptomic variability through density-preserving data visualization.* Nature Biotechnology, 39(6):765–774, 2021.

[4] Kobak, D, Berens, P. *The art of using t-sne for single-cell transcriptomics.* Nature Communications, 10(1):1–14, 2019.

[5] Kury, N et al., *DREAMS: Preserving both Local and Global Structure in Dimensionality Reduction*, arXiv:2508.13747, 2025.

**Questions:**

-	You claim that through the diffusion you get both local as well as global properties of the data in the embedding. How much does your **choice of considering the kNN graph influence the results which were derived without that constraint**?
-	How do you get the stationary distribution $\pi_Q$?
-	The final **optimization objective (Eq. 13) looks eerily close to the tSNE objective**, with $Q_{ij}$ even being a kernel functions using l2 distances the same way that tSNE does. Could you elaborate what the difference is?
-	I know the student t kernel. **What is the student -t kernel?**
-	What are the results for **each method (including yours and above mentioned SOTA), each metric (not aggregated across metrics) and each dataset (not aggregated across datasets)**? Almost all of this is missing in the paper.

---

### Official Review · Reviewer_pzep · 2025-11-01

**Soundness:** 3
**Presentation:** 3
**Contribution:** 2
**Rating:** 2
**Confidence:** 4

**Summary:**

This paper introduces Diffusion-Aligned Embeddings (DAE), a novel diffusion-based dimensionality reduction method that aligns diffusion processes between high- and low-dimensional spaces. The key idea is to use the Path-KL divergence to align diffusion generators. By minimizing this divergence, DAE formally guarantees closeness between the high and low dimensional diffusion semigroups across scales. The authors derive theoretical bounds on generator and semigroup preservation and propose an efficient, parallelizable optimization algorithm with unbiased stochastic gradients.

**Strengths:**

The paper presents a rigorous formulation of dimensionality reduction based on aligning diffusion generators, rather than fixing a single diffusion timescale. The use of the Path-KL divergence is a novel and elegant idea that provides formal multiscale preservation guarantees and removes the need to choose a specific diffusion time parameter t, a common limitation in methods like PHATE and diffusion maps.  The theoretical analysis is technically sound and clearly motivated, connecting generator alignment to semigroup closeness with well-defined mathematical guarantees. The proposed optimization framework is well-engineered and scalable, demonstrating practical feasibility for large datasets.

**Weaknesses:**

Second surprising fact is that the the authors provide no visualizations in the main text, making it difficult to assess whether the embeddings preserve meaningful global geometry or suffer from distortions. Moreover the metrics they use are obscure. They should use manifold affinity preservation as in PHATE.

The embedding looks somewhat similar to tSNE from the one figure in the appendix. The reason for this is not surprising, tSNE also does a limited version of matching a diffusion process. Conceptually, matching diffusion processes via generator alignment may not guarantee geometric faithfulness. Because diffusion probabilities are locally normalized, embeddings that differ by large-scale rescaling or local contraction/expansion can produce equivalent diffusion behavior. Thus, the method can preserve relative diffusion dynamics while arbitrarily distorting the manifold geometry.

The paper omits direct comparison to methods like PHATE, multiscale PHATE and diffusion maps, despite positioning itself as a diffusion-based embedding method. This omission is surprising, as multiscale PHATE also models multiscale diffusion structure and is a clear conceptual predecessor. The authors should include both in the related work and the additional experimental comparisons. Another point of comparison would be the Heatgeo embedding [Huguet et al. NeurIPS 2023] which unifies diffusion based processes like PHATE, tSNE and diffusion maps and diffusion maps.

.
The paper would benefit from a dedicated background section explaining key concepts such as diffusion maps, diffusion distances, and continuous-time Markov chains before introducing the Path-KL objective. Important prior work like PHATE (and variations) and diffusion maps should be summarized in a background or related work section to clarify how DAE builds on or differs from existing diffusion-based methods. The current exposition moves quickly into derivations without establishing sufficient mathematical or conceptual background, making it difficult for readers unfamiliar with diffusion geometry to follow. Figure 2 should be in the main body of the paper. Also should include comparisons to PHATE.

**Questions:**

How do you ensure that preserving the diffusion process also preserves geometry?

Why dont you show comparisons to other diffusion based methods?

---

### Official Review · Reviewer_8ypj · 2025-11-01

**Soundness:** 3
**Presentation:** 2
**Contribution:** 2
**Rating:** 4
**Confidence:** 2

**Summary:**

This paper explores a new representation learning framework that models data as a stochastic diffusion or random walk process, formulated through a continuous-time Markov chain (CTMC). The authors aim to capture intrinsic relationships among data points beyond static embeddings by treating representation learning as a dynamic probabilistic evolution over time. Overall, the paper presents a clear narrative and the proposed framework is intuitively appealing. However, the study appears insufficiently mature for ICLR. While the topic is conceptually relevant, there are notable issues with scope, technical depth, and experimental completeness that weaken its overall contribution.

**Strengths:**

S1. The paper is logically structured, with a clear narrative and motivation.

S2. The proposed problem is meaningful and lies within the broader scope.

S3. The writing is fluent and the paper is easy to read.

**Weaknesses:**

**Concerns**

C1. While representation learning is indeed a core theme at ICLR, this paper appears to align more closely with scientific or interdisciplinary journals rather than a machine learning conference. The reference list includes almost exclusively Nature or Science publications, with no ICLR or ICML papers cited. This suggests a lack of engagement with the ML research community. Furthermore, the most recent related work in Section 1 cited dates back to 2021, raising concerns either about insufficient literature review or about the maturity and saturation of the studied problem. The authors are expected to better clarify how this work advances the state of machine learning beyond existing literature.

C2. Although the paper’s ideas are clearly articulated, the technical core appears relatively shallow. The proposed method resembles a random-walk process enhanced with continuous-time Markov modeling, but lacks substantial theoretical innovation or algorithmic novelty. It would be better if the authors emphasize what makes their formulation non-trivial—for example, any novel mathematical insights, optimization challenges, or new theoretical guarantees. Otherwise, the approach risks being perceived as an incremental reformulation of well-known stochastic models.

C3. The experiments are currently insufficient to convincingly support the paper’s claims. It would better if author can address these issues to significantly improve the paper’s credibility and completeness:

C3-1. The experimental design is limited in diversity—more datasets or settings would strengthen generality.

C3-2. The evaluation metrics are not clearly explained.

C3-3. The discussion of results is brief or missing, lacking qualitative or interpretive analysis that explains why the proposed method performs as it does.

C3-4. Ablation studies or sensitivity analyses could help clarify the contribution of individual components.

C3-5. No runtime or complexity analysis is provided to demonstrate the practicality of the approach.

**Questions:**

Please respond to C1, C2, and C3.

---

### Official Review · Reviewer_FT2T · 2025-11-03

**Soundness:** 3
**Presentation:** 2
**Contribution:** 2
**Rating:** 4
**Confidence:** 4

**Summary:**

This paper proposes a new method for dimensionality reduction that relies on matching the generator matrices of CTMC built on (1) a graph built upon the high dimensional data and (2) a graph built upon the low-dimensional data.
A key insight of this paper is to minimize the KL divergence between the path distributions (equivalent to the relative entropy rate), which should enable multi-scale representation of the data (because it's matching these distributions at all times t). The paper also proposes an efficient algorithm to compute the embeddings.

The authors compare the performance of their method on 5 different datasets, although it's not clear results of all datasets are actually presented in the paper.

**Strengths:**

- This paper proposes an interesting approach to avoid the problem of choosing a diffusion time (which is an important hyper-parameters in diffusion-based dimensionality reduction methods).
- The authors designed an efficient computational algorithm to compute the embeddings, enabling it to scale to millions of data points (225 seconds for 9.5 M points in 50 dimensions).

**Weaknesses:**

- Although most of Section 2 is well presented and easy to follow, the authors fail to nail down the critical equivalence between Eq (4) and Eqs (6-7). If I understand correctly, this is the key of this paper, as Eq 4 gives the general motivation and 6-7 provides a computationally tractable way of minimizing that objective.
- Q is effectively the Laplacian of the graph, that should be stated in the text.
- Another recent work also uses heat-diffusion on a graph for dimensionality reduction [1]. In particular, that work showed that directly using the heat kernel (the matrix exponential of the Laplacian), already corresponds to combining diffusion operators at multiple scales. However, this is not discussed in the paper.
- The experiment section should contain more results in the main text. The text suggests there are 5 datasets but Figure 1 seems to only be on a single dataset with different values of k. It's not even clear what dataset this refers to. I encourage authors to clarify the results.
- Also in the results, you should report DAE as its own bar, as the current presentation erases the stochasticity of the method. You should also not normalize to 1, such that we can appreciate the difference in performance across multiple k. In particular, it's pointless to improve upon t-SNE for k<30 if absolute performance are worse in that regime than for k>30.
- As per the presented results, there is no convincing evidence that their method is better than t-SNE.


[1] Huguet, Guillaume, et al. "A heat diffusion perspective on geodesic preserving dimensionality reduction." Advances in Neural Information Processing Systems 36 (2023): 6986-7016.

**Questions:**

- Could you please clarify the link between  between Eq (4) and Eqs (6-7), as I pointed to above ?
- Could you elaborate on the connection with the work in [1], as referred above?
- Could you incorporate more quantitative (and at least one convincing qualitative) results in the main text ?
- Figure 1 should be refactored completely. You should report DAE as its own bar, as the current presentation erases the stochasticity of the method. You should also not normalize to 1, such that we can appreciate the difference in performance across multiple k. In particular, it's pointless to improve upon t-SNE for k<30 if absolute performance are worse in that regime than for k>30.
- If possible, it would be great to show a dataset where DAE outperforms tSNE.
- An important contribution of this paper is the efficient algorithm for computing the embeddings. However, there is no results comparing the performance of different methods. I encourage the authors to provide such comparisons.

---

### Note · Authors · 2025-12-04

**Comment:**

We thank the reviewers for their detailed and thoughtful feedback. After reflecting on the comments, we have decided to withdraw this submission and substantially revise it.

At a high level, our aim in DAE is to formulate dimensionality reduction as aligning continuous-time diffusion processes between the high- and low-dimensional graphs, using the CTMC path-space relative entropy rate (Path-KL) as the objective. This quantity captures both which neighbors are chosen and how long the process waits at each node via the generator $\mathcal{Q}$, and the paper proves that small Path-KL implies generator closeness and semigroup closeness across timescales, rather than committing to a single diffusion time $t$.

Several of the reviewers’ concerns stem from presentation choices, and we agree that the current draft does not communicate the technical core and empirical evidence as clearly as it should. In particular:

(1) The link from the path-KL objective in Eqs. (4–5) to the decomposition in Eqs. (6–7) is the standard relative entropy rate formula for CTMCs; Lemma 2.1 states the closed form and Appendix A contains a step-by-step derivation (Eqs. 14–18). We agree this deserves a short derivation sketch and discussion in the main text, rather than being entirely deferred to the appendix.

(2) The paper currently assumes an irreducible (ergodic) generator in the theoretical results to ensure a unique stationary distribution and a well-defined entropy rate, which is standard in this setting. In our implementation we build a symmetrized k-NN graph with symmetric rates, so the generator is reversible and the stationary distribution has a simple closed form (uniform in this symmetric-rate case). We will make this practical/theoretical alignment explicit in the revision, and clarify that reversibility is a modeling choice for implementation, not a requirement of the Path-KL objective itself.

(3) PHATE is included as a baseline in the experiments, but we agree that the related-work section and discussion should better situate DAE relative to diffusion maps, PHATE/multiscale PHATE, and recent heat-kernel / HeatGeo-style methods, as several reviewers noted.

On the experimental side, we agree that the presentation is confusing and incomplete in its current form. Figure 1 reports ratios to a baseline (with a red line at 1.0) across datasets and metrics, which can obscure absolute performance and variability and can make the baseline method visually disappear. Additional per-dataset quantitative results and qualitative embeddings are currently pushed to the appendix. In a revised version, we plan to (i) move representative qualitative embeddings into the main text, (ii) report per-dataset, per-metric tables and plots with error bars, (iii) include more explicit runtime and scaling comparisons, and (iv) add ablations/sensitivity analyses over key hyperparameters (kernel, bandwidth, α negatives, etc.), as requested by multiple reviewers.

We appreciate the feedback and will use it to substantially restructure and clarify the next version, both in terms of exposition (making the Path-KL / diffusion-alignment perspective and assumptions more transparent) and in terms of experimental completeness and positioning.

**Withdrawal Confirmation:**

I have read and agree with the venue's withdrawal policy on behalf of myself and my co-authors.